# Influence of Different Orthodontic Brackets on Cytokine and Cortisol Profile

**DOI:** 10.3390/medicina59030566

**Published:** 2023-03-14

**Authors:** I. Pantsulaia, N. Orjonikidze, I. Kvachadze, T. Mikadze, T. Chikovani

**Affiliations:** 1Department of Immunology, Tbilisi State Medical University, Tbilisi 0177, Georgia; 2Vl. Bakhutashvili Institute of Medical Biotechnology, Tbilisi State Medical University, Tbilisi 0159, Georgia; 3Department of Orthodontics, Tbilisi State Medical University, Tbilisi 0177, Georgia; 4Dental Clinic and Training-Research Center “Unident”, Tbilisi 0177, Georgia; 5Department of Physiology, Tbilisi State Medical University, Tbilisi 0177, Georgia

**Keywords:** cytokine, self-ligating brackets, conventional brackets

## Abstract

*Background and Objectives*: Orthodontic tooth movement (OTM) requires bone remodeling resulting from complex processes of aseptic inflammation. Recent studies have confirmed close interaction between the immune and skeletal systems. In addition, various orthodontic appliances including fixed systems affect the sublingual microbial composition, and the likelihood of developing inflammatory reactions of the gums is high, especially early in the treatment period. It is known that these systems have both positive and negative effects on the humoral and cellular immune responses. The main aim of the study was to evaluate the influence of self-ligating and conventional brackets on the salivary concentrations of cytokines (IL-6, osteoprotegerin (OPG), TNF-alpha, and IFN-gamma) and cortisol as a marker of stress. *Materials and Methods*: Forty patients were analyzed at baseline (T0) and 2 months (T2) after fixing self-ligating (Ormco Damond Q) and conventional brackets (Ormco Mini Diamond). Salivary cytokine and cortisol concentrations were evaluated by commercial ELISA kits. *Results*: Outcomes of our study showed that after two months of treatment with either of these brackets, IFN-gamma and IL-6 levels did not change. However, TNF-alpha decreased with self-ligating brackets (13.36 to 8.32, *p* = 0.002). The self-ligating bracket system also affects OPG concentration and cortisol levels 2 months after orthodontic activation. The level of OPG in the group of self-ligating brackets decreased significantly (8.55 to 2.72, *p* = 0.003). Cortisol concentration was significantly higher in the self-ligation group (25.72 to 48.45, *p* = 0.001) due to the effect of sustained strength movements. *Conclusions*: Thus, the use of self-ligating and conventional brackets has a different effect on the concentration of cortisol and cytokines (OPG and TNF-alpha) in saliva 2 months after their fixation. Further longitudinal studies are necessary to explore why OPG levels are decreased in case of self-ligating cases and how OPG levels are related to clinical improvement.

## 1. Introduction

Orthodontic tooth movement (OTM) requires bone remodeling, which results from complex processes of aseptic inflammation. Recent works have confirmed a close relationship between the immune and skeletal systems [1,2,3,4,5]. Studies have shown that bone remodeling involves osteoclasts’ (OC) and osteoblasts’ (OB) differentiation and activation, mediated by immune system [1,6,7,8]. The cytokines are responsible for change in the cytoskeletal structure, leading to alteration of the nuclear protein matrix and eventually gene activation or suppression. Formation of osteoclasts occurs in two “waves”. The first wave is generated by the exposure of local population of cells to cytokines produced by periodontal ligament fibroblasts (PDLF) in response to mechanical stress [4,8,9]. The second phase of osteoclast formation originates from leukocytes extravasated from dilated blood vessels [10] migrating into the tissue and secreting anti-inflammatory cytokines, chemokines, growth factors, and enzymes [11]. 

Specific mediators associated with compression and stretching zones regulate the processes of bone resorption and deposition. Compression is associated with an increase in cycloxygenase-2 (COX-2). It produces prostaglandins from arachidonic acid, including PGE2 (prostaglandin) [12]. Prostaglandins act on osteoclasts, increase intracellular Ca concentration, and enhance their resorption activity. It has been suggested that the PDLF is the first cell to initiate molecular response to orthodontic forces, that is, production of osteoclastogenesis promoters such as PGE2 and RANKL in response to compressive force. PGE2 stimulates differentiation of osteoblasts and expression of RANKL and OPG [12]. The elevation of RANKL (receptor activator of nuclear factor κ B-ligand) and M-CSF and reduction of osteocytes jointly serve to differentiate osteoclasts and bone resorption. Additionally, proinflammatory cytokines (IL-1B and TNF-alpha) lead to differentiation of osteoclasts; enhance their function, survival, and inflammation; and increase matrix-metalloproteinase levels [12,13]. Compression also activates nitric oxide synthase (iNOS), which produces nitric oxide (NO), leading to inflammatory bone resorption [14]. These factors attract and activate osteoclasts that produce resorption lacunas in the compressive zone. Movement of the teeth begins after the necrotic tissue is removed by osteoclasts. Thereafter, osteoblasts form osteoid, i.e., new periodontal fibrils, which are inserted into the alveolar bone wall and root cement. Expression of bone morphogenetic proteins (BMPs) and Runx2 enhances differentiation of osteoblasts and bone mineralization, while proliferating and active fibroblasts enhance the formation of extracellular matrix fibers [12]. Compressed bone and PDL are destroyed and regenerated again. Stretch in the PDL stimulates proliferation of osteoblast progenitors and activates endothelial nitric oxide synthase (eNOS), which promotes bone formation through NO production [14]. In stretching areas, osteoprotegerin (OPG) levels as well as IL-10 production is increased, and production of RANKL by osteoblasts is reduced. Downregulation of receptor nuclear factor kappa (RANK) signaling by inhibiting osteoclast formation, activity, and survival promotes bone deposition [15]. Furthermore, the amount of TGF-b elevates, proliferation and chemotaxis of PDL cells occur, the collagen gene (COL-I) is expressed, osteoblast precursors are attracted and differentiated, the amount of matrix metalloproteinases (MMPs) is decreased, and tissue inhibitors of metalloproteinases (TIMP) are increased [9,16]. As a result of increased activity of osteoblasts and decreased activity of osteoclasts, bone formation and remodeling of PDL fibers occur on the opposite side of tooth movement [17,18]. The expression OPG and Runx2 in response to stretch is also increased by vitamin D3, PTH, IL-1, IL-6, IL-8, IL-11, IL-17, IL-32, and TNF. Most of these molecules are cytokines produced by the cells of the immune system. IL-4, IL-10, IL-18, and IFN have the opposite effect. They inhibit osteoclastogenesis [19]. Thus, the activation of nuclear factor kappa-ligand (RANKL)-nuclear factor kappa-(RANK)-osteoprotegerin (OPG) interactions is a major bone remodeling pathway associated with tooth movement [15]. 

Recently, approximately 3075 differentially expressed genes were identified, whose products are involved in this process [20]. Klein et al. [20] observed two different pictures: one where the maximum gene expression occurred in the first days after fixation with subsequent decrease (tissue degradation and innate and adaptive immune response) and the second where gene expression was initially suppressed followed by activation on day 14 (cell proliferation and migration, cytoskeleton rearrangement, cellular homeostasis, and angiogenesis). The discovery of new immune mechanisms related to forceful movement of teeth led researchers to introduce a new term: “immunorthodontia” [20].

Thus, it appears that the inflammatory cascade is critical for orthodontic tooth movement. However, dysregulated or excessive inflammation is always problematic for the body. Orthodontically induced tissue remodeling should only affect bone and periodontal tissues, not the cementum and teeth. However, in 1–5% of orthodontic patients, excessive root resorption is observed, with a loss greater than 4 mm [21,22]. Root-length shortening reduces the crown-to-root ratio of affected teeth, which is potentially of great clinical importance. These studies demonstrate that the orthodontic force application engages the cellular and molecular pathways of inflammation. However, the exact biological factors, processes, and time of development are still unclear. Therefore, the main aim of this study was to evaluate the parameters of oral immune homeostasis when using various orthodontic brace systems (conventional and self-ligating). 

## 2. Materials and Methods

Subjects: A G-POWER analysis (alpha 0.05, power 0.80, large effect size (d = 0.8), and two-tailed) was performed to determine the minimum number of study subjects. Based on the aforementioned assumptions, the desired sample size was 15. Our preliminary and observational study was conducted on healthy individuals (aged 13–24 years) who underwent orthodontic treatment with fixed brackets. During the study period (2021), 80 patients started orthodontic treatment in the Dental clinic and Training-Research Center “Unident”. Out of these, 50 met the inclusion criteria and were distributed into the two groups (I, conventional brackets (CB); II, self-ligating brackets (SB)). Inclusion criteria were dentition abnormalities of the patients: deep dentition, open dentition, prognathia, progenia, and cross dentition. The presence or coexistence of other types of orthodontic pathology was an exclusion criterion. Additionally, users of immunosuppressive drugs or alcohols and/or individuals with pathologies affecting the immune system (infectious diseases, tumors, autoimmune and inflammatory pathologies, chronic liver pathologies, diabetes, and asthma) were excluded from the study. Information about 10 patients was withdrawn from the final data because of acute respiratory diseases (COVID-19, flu, etc.) during observation period. Thus, 25 patients in the CB group and 15 in the SB group were eventually analyzed. Data on age, gender, sexually transmitted infectious and chronic diseases, and medical treatment were collected for each individual using a special questionnaire. The study was performed in the Department of Orthodontics, Dental Clinic and Training-Research Center “Unident” and the Immunorehabilitation Laboratory of the Vl. Bakhutashvili Institute of Medical Biotechnology at Tbilisi State Medical University. The study was performed in accordance with the terms of the Helsinki Ethics Commission. All the procedures envisaged in the study were approved by the Bioethics Commission of Tbilisi State Medical University. Each individual’s participation was voluntary and confirmed by the signature on the questionnaire.

### 2.1. Collection of Saliva

Saliva was collected between 9 a.m. and 12 a.m. at least one hour after eating and oral hygiene to reduce the effect of diurnal variation in saliva composition. Saliva was collected in 2.0 mL sterile containers and stored frozen at −20 °C. Unstimulated saliva samples were collected 30 min before treatment (T0, before fixation of the orthodontic fixed system) and 2 months after treatment (T2).

### 2.2. Clinical Assessment of Oral Cavity

Before saliva collection, the clinical condition of the periodontium was evaluated by one person according to plaque index (PI) and bleeding on probing (BOP). Degree of crowding was similar; in all cases, it was moderate crowding. Plaque was quantified according to Silness and Loe’s criteria [19]. The plaque index was calculated using a standard probe and direct observation. The amount of plaque was evaluated with four points: 0—no plaque, 1—plaque attached to the free edge of the gum and surrounding area of a tooth that can be seen on the surface of a tooth with a probe, 2—moderate amount of a soft plaque visible to the naked eye in the gingival pocket or at the tooth/gingival margin, 3—large amount of a soft plaque visible in the gingival pocket and/or at the tooth/gingival margin. Six sites were analyzed for each tooth (mesiobuccal, buccal, distobuccal, distolingual, lingual, and mesiolingual). Plaque index (PI) was determined by the percentage of a tooth surface with plaque.

For assessment of the gingiva, we used bleeding on probing (BOP) according to Loe and Silness [19]. A BOP is considered positive if it occurs within 20 s of probing. The result is evaluated from 0 to 3 points: 0 = healthy gingiva; 1 = mild inflammation: slight color change, slight swelling, and no bleeding on probing; 2 = moderate inflammation: redness, swelling, and bleeding on probing; 3 = severe inflammation: pronounced redness and swelling, tendency to spontaneous bleeding, and presence of ulcers. Only patients with PI and BOP index 0 or 1 were included in the study.

### 2.3. Cytokine Measurement

The cytokines (IL-6, TNF-alpha, osteoprotegerin (OPG), and interferon gamma (IFNγ)) and cortisol were evaluated by means of commercial immunoenzymatic (ELIZA) kits (Mybiosources Inc., San Diego, CA, USA). Briefly, the kits were analytically validated with ready-to-use reagents. To measure cytokines and cortisol, standards and samples were added to the wells, then the biotinylated detection antibody was added. The wells were washed with PBS or TBS buffer, and Avidin–Biotin–Peroxidase Complex (ABC-HRP) was added. The absorbance of the yellow product in each well was read using a plate reader (BioTek, Synergy HTX, Agilent, Santa clara, CA, 95051, United States). All measurements were performed in pairs, the mean was used, and coefficients of variation between analysis were estimated. The sensitivity and specificity of the test had to exceed 97%. When selecting sets, it was considered that the coefficients of variation between the samples were not more than 5% and 10%.

### 2.4. Statistical Analysis

Statistical calculations were performed by means of the STATISTICA (Stat soft, Inc, USA). The general strategy of the quantitative traits analysis included the following main stages: (1) preliminary descriptive statistics and (2) statistically independent groups (factors) of the dependent variables were created. 

The normal distribution of the cytokines and cortisol was assessed by the Kolmogorov–Smirnov and Lilliefors test for normality. Since the distributions of the studied molecules were markedly skewed, the data on biochemical markers were log-transformed to correct for non-normality prior to further analysis. Review of the distribution properties characterizing the studied molecules showed some of the values to be outliers, that is, inconsistent with the distribution of the bulk of the data. To ensure that some rare observations were not ultimately excluded, we used ≥4 SD as the range criterion for outliers; all the relevant saliva samples were then re-assayed before final exclusion from further analysis. Differences in means of the plasma cytokines between male and female groups were determined by the Mann–Whitney U test. Additionally, the median and interquartile ranges for each studied parameter were calculated and compared by non-parametric analysis. The original values are presented by mean, standard deviations, minimum and maximum, and median with interquartile ranges (Table 1, Table 2 and Table 3). Two-way repeated measures ANOVA was used at specific time points to evaluate differences between the groups. A *p*-value of 0.05 or less was deemed significant for all analyses. 

## 3. Results

The baseline characteristics and clinical features of the patients in the CB and SB groups are shown in Table 1. Data on cytokines and cortisol are presented before log-transformation and in the original units. At the first stage of analysis, the investigated cytokines (OPG, TNF-alpha, IFN-gamma, and IL-6) and cortisol concentrations were scanned by gender and age. The span of variation of all the variables in this study was within the range of normal variation for each gender (Table 1). A comparison of the mean levels did not reveal marked gender dependency (Mann–Whitney U test, Table 1). Since the values for men and women were not significantly different, further statistical analysis was continued using combined data. Next, correlations between the age and cytokines as well as cortisol were calculated, and it was found that none of the cytokine levels depended on age (*p* > 0.05). Furthermore, the baseline mean values for CB and SB groups were compared, and none of the cytokine levels showed statistically significant differences at that time point (Table 1).

The levels of OPG and TNF-alpha in the saliva of patients with conventional brackets did not show a statistically significant change 2 months after the start of the treatment (Table 2). 

However, in case of using self-ligating brackets, the concentrations of OPG and TNF-alpha significantly decreased two months after the start of the treatment (Figure 1 and Figure 2). Moreover, the level of OPG was significantly lower 2 months after fixation of self-ligating brackets than in case of conventional brackets (Figure 1, *p* = 0.003).

Two months after fixation of self-ligating brackets, the IL-6 level decreased, although the decrease is not statistically credible (Table 3). It is possible that this is due to very fluctuating levels of interleukin-6 before fixation. In case of conventional brackets, the level of IL-6 did not change (13.18 to 19.94; *p* = 0.67, Table 2). 

It appears that two months after the start of the treatment, there was significant difference in the salivary IL-6 levels between patients treated with conventional and self-ligating brackets: two months after fixing self-ligating brackets, it was lower than in the case of conventional brackets, but this difference is not statistically significant (10.94 ± 13.60 and 4.99 ± 5.00, respectively; *p* = 0.11) (Figure 3). As regards IFN-gamma, its concentration was not altered according to the investigation group (Table 3).

Because the orthodontic movement is associated with the patient’s stress, we also evaluated the saliva level of cortisol in the studied groups (Table 2 and Table 3). The analysis of mean values at different time points showed that 2 months after treatment with conventional brackets, cortisol levels were not different from the baseline level (*p* = 0.90). As regards the self-ligating bracket group, cortisol levels were statistically significantly increased 2 months after starting the treatment (Figure 4, *p* = 0.01).

It has to be noted that comparing the mean values of studied groups revealed a statistically significant difference between them (Figure 4, *p* < 0.01). 

## 4. Discussion

Improving the quality, speed, and stability of orthodontic tooth movement is crucial for practical medicine. It is well known that pro- and anti-inflammatory cytokines are involved in the regulation of differentiation and activation of osteoclasts and osteoblasts, which ensures the maintenance of bone homeostasis, especially in response to aggressive external agents [23,24,25,26,27]. Moreover, the expression of inflammatory cytokines plays a crucial role in orthodontic movement, as they increase the safety of fixation systems with sustainable suspension [18,28]. Inflammatory processes are central during orthodontic movements, although the exact mechanism is unclear. It is especially important to assess whether significant differences in cytokine levels occur with various fixation systems and how much this is reflected in the shortening of treatment duration and in the reduction of complications.

Based on our findings, after 2 months of starting treatment with conventional and self-ligating brackets, the concentration of IFN-gamma in saliva did not change compared to baseline data. However, 2 months after fixation, the patients’ salivary osteoprotegerin concentrations were statistically significantly lower in the case of self-ligating brackets. Furthermore, according to the results of our research, bone resorption was in an active phase even after 2 months of using the self-ligating brackets. The process is time-dependent [29]. This can be explained by the biodegradation and relaxation of elastomers when using conventional brackets. Our results agree with Mukherjee et al. [30], who indicated that OPG concentrations increased at 48 h, and OPG levels were up-regulated during intermittent stress. Kusumi et al. [31] found that OPG levels change after applying tensile stress to human osteoblasts. However, Florez-Moreno et al. [32] made the opposite finding and reported that OPG levels decreased during mechanical loading (24–48 h). Thus, we can speculate that in the case of self-ligation, the diminished levels of OPG are associated with ischemia and hypoxia due to mechanical compression of the microvasculature.

The self-ligating bracket system is known to have reduced friction [33], more mechanisms of action and less treatment time, increased patient demand, and higher-quality outcomes. However, some studies indicate [34] that there are no significant differences in long-term outcomes between self-ligating and conventional brackets.

Orthodontic treatment is also accompanied by stress [35]. Stress is a combination of physiological and psychological reactions that protect the body from external or internal stressors by increasing cortisol secretion [36]. According to the results of our study, the cortisol level was not altered after fixing conventional brackets, although it increased significantly after using self-ligating brackets (Table 3). Other authors also found that changes in salivary cortisol during treatment depended on the type of orthodontic brackets [37]. Our results are in full agreement with the results obtained by Andrade and co-authors [38] 24 h and 30 days after the start of the treatment. However, they contradict the data of Chetan et al. [39], who found that there is a statistically significant decrease in salivary cortisol levels in orthodontic patients 1 h and 4–6 weeks after the start of the treatment. Moreover, Canigur et al. [40] showed that orthodontic tooth movement did not cause significant alterations in salivary pain and stress biomarkers.

We also found that IL-6 and TNF-alpha concentrations were different between those using conventional and self-ligating brackets after 2 months of treatment. As pro-inflammatory cytokines, they indicate the activation of resorption during bone remodeling. According to Andrucioli et al. [41], IL-6 levels are significantly higher in failed mini-implants. However, Andrucioli et al. [41] observed no differences between the groups regarding RANK, RANKL, and OPG levels. Duarte et al. [42] demonstrated higher OPG expression in the group with more severe inflammation (peri-implantitis). Our results partially agree with Bergamo et al. [27], who showed that bracket design affects salivary cytokine concentrations and should be considered as a risk factor for patients with periodontitis. Bergamo et al. [27] revealed that the SmartClip™ group brackets increase the level of TNF-alpha. This may be related to a high bacterial level, which has greater inflammatory potential. Thus, based on our results as well as previous studies, we can consider that the inflammatory stimuli are lower in the case of self-ligation compared to conventional brackets.

Although our results are convincing, this study had some limitations. First, evaluation of the treatment efficacy in our study model is based on prospective observational studies rather than randomization. Second, the sample size is small; the number of various groups (CB—25 and SB—15, respectively) is different; and third, this is a single-center study, and its results should be treated with caution as general and apparently valid.

## 5. Conclusions

In conclusion, it can be emphasized that self-ligating and conventional brackets have different effects on the concentration of cytokines (OPG, TNF-alpha, and IL-6) and cortisol in saliva 2 months after their fixation.

Further long-term studies with time intervals are needed to accurately assess what accounts for the difference in cortisol and cytokine concentrations between these two different groups. It is also important to measure the cytokine concentrations not only in saliva but also in the space between the fixation systems and the bone tissue. It will facilitate new approaches, particularly the development of safe, accelerated dental orthodontic techniques.

## Figures and Tables

**Figure 1 medicina-59-00566-f001:**
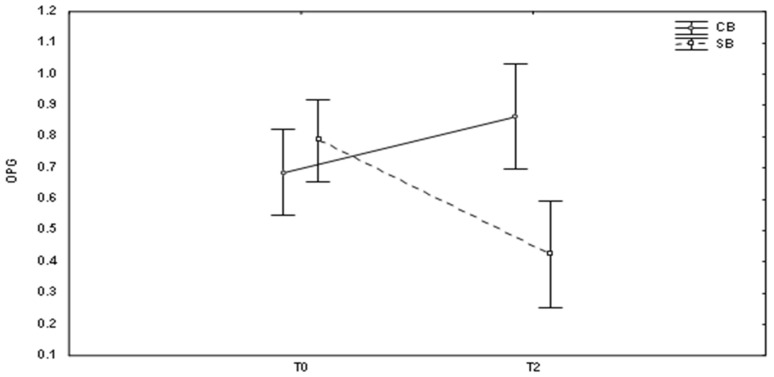
Changes in OPG (mean ± SD) levels at different times (T0, baseline; T2, after 2 months of the treatment) in patients with conventional brackets (CB) and self-ligating brackets (SB).

**Figure 2 medicina-59-00566-f002:**
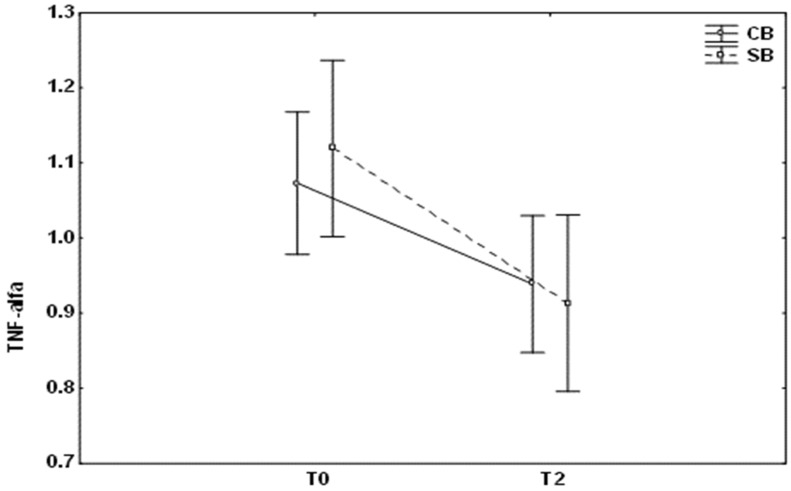
Changes in TNF-alpha (mean ± SD) levels at different times (T0, baseline; T2, after 2 months of the treatment) in patients with conventional brackets (CB) and self-ligating brackets (SB).

**Figure 3 medicina-59-00566-f003:**
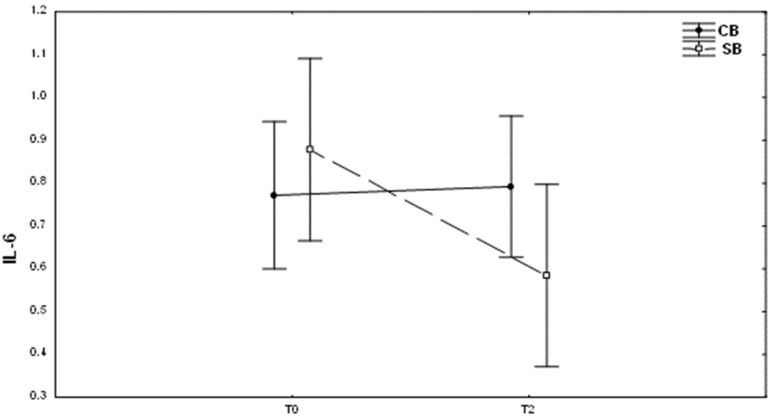
Changes in IL-6 (mean ± SD) levels at different times (T0, baseline; T2, after 2 months of the treatment) in patients with conventional brackets (CB) and self-ligating brackets (SB).

**Figure 4 medicina-59-00566-f004:**
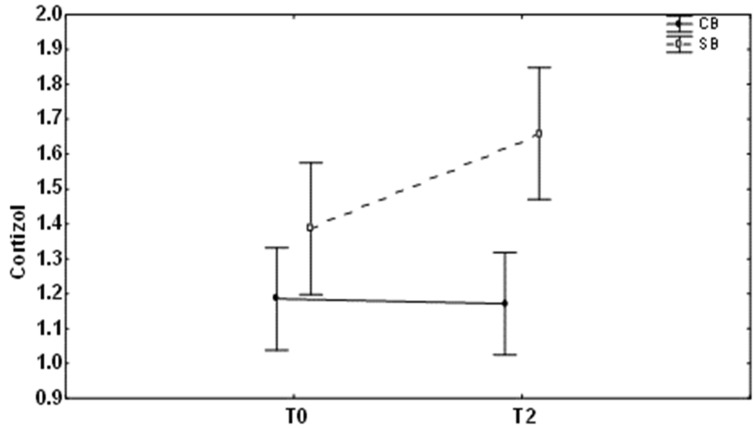
Changes in cortisol (mean ± SD) levels at different times (T0, baseline; T2, after 2 months of the treatment) in patients with conventional brackets (CB) and self-ligating brackets (SB).

**Table 1 medicina-59-00566-t001:** The baseline characteristics (mean ± std.dev.) of the studied traits in various groups according to gender: I, conventional brackets (CB); II, self-ligating brackets (SB).

I CB	Mean ± Std.Dev.			
	Total	Men	Women	*p*
Age	16.78 ± 3.56	17.00 ± 4.43	16.53 ± 3.22	0.78
Cortisol	23,19 ± 19,23	23.75 ± 25.55	23.01 ± 17.65	0.94
OPG	7.03 ± 6.57	6.42 ± 5.36	7.23 ± 7.02	0.80
TNF-alpha	13.49 ± 8.14	16.29 ± 9.53	12.50 ± 7.67	0.34
IL-6	13.18 ± 25.17	16.23 ± 18.42	12.11 ± 27.58	0.74
IFN-gamma	7.97 ± 5.41	6.98 ± 1.58	8.28 ± 6.16	0.62
II SB	Mean ± Std.Dev.			
	Total	Men	Women	*p*
Age	15.93 ± 3.42	17.4 ± 4.16	16.2 ± 3.70	0.58
Cortisol	25.720 ± 9.11	22.59 ± 6.56	27.29 ± 10.08	0.37
OPG	8.55 ± 5.33	9.13 ± 6.16	8.26 ± 5.20	0.78
TNF-alpha	13.36 ± 2.42	12.18 ± 2.29	13.94 ± 2.37	0.20
IL-6	13.06 ± 20.16	20.27 ± 33.55	9.46 ± 9.40	0.35
IFN-gamma	8.84 ± 5.83	9.44 ± 6.91	8.54 ± 5.60	0.80

**Table 2 medicina-59-00566-t002:** Dynamics of salivary cytokine levels (mean ± std.dev (min–max), median (25–75%), and confidence intervals 95%) in patients treated with conventional brackets (CB).

	Mean ± Std.Dev (Min–Max); Median (25–75%) Confidence Intervals 95%		*p* (T0–T2)
	T0	T2	
Cortisol	23.19 ± 19.23 (1.06–69.68) 15.76 (8.81–36.99) 15.25–31.13	22.44 ± 18.82 (1.64–63.83) 14. 60 (11.86–34.76) 14.67–30.21	0.90
OPG	7.03 ± 6.57 (2.28–11.70) 4.66 (2.04–25.00) 2.04–25.00	9.93 ± 10.59 (1.48–41.30) 5.51 (2.50–11.14) 5.56–14.30	0.42
TNF-alpha	13.49 ± 8.14 (6.93–37.50) 10.38 (7.90–16.09) 9.96–17.01	11.89 ± 11.40 (4.92–54.63) 9.06 (6.40–11.59) 7.18–16.59	0.10
IL-6	13.18 ± 25.17 (2.42–116.27) 3.87 (2.552–9.05) 2.30–24.07	10.94 ± 13.60 (2.45–52.06) 3.76 (2.84–17.73) 5.33–16.56	0.67
IFN-gamma	7.97 ± 5.41 (4.94–27.40) 6.01 (5.78–7.33) 5.74–10.20	9.56 ± 14.00 (4.78–65.00) 5.67 (2.84–17.73) 3.78–15.33	0.25

OPG, osteoprotegerin; TNF, alpha-tumor necrosis factor alpha; IL-6, interleukin 6; IFN, gamma-interferon gamma; T0, before treatment; T2, two months after fixing the brackets. Indicators were statistically significantly different when *p* < 0.05.

**Table 3 medicina-59-00566-t003:** Dynamics of salivary cytokine levels (mean ± std.dev (min–max), median (25–75%), and confidence intervals 95%) in patients treated with self-ligating brackets (SB).

	Mean ± Std.Dev (Min–Max); Median (25–75%) Confidence Intervals 95%		*p* (T0–T2)
	T0	T2	
Cortisol	25.72 ± 9.11 (15.22–42.50) 25.61 (18.56–32.510 20.678–30.762	48.45 ± 18.11 (31.14–78.45) 38.92 (32.80–69.45) (38.43–58.48)	0.001
OPG	8.55 ± 5.33 (3.33–18.07) 5.88 (5.13–15.26) 5.60–11.50	2.72 ± 0.73 (2.06–5.24) 2.58 (2.45–2.78) 2.32–3.13	0.003
TNF-alpha	13.36 ± 2.42 (10.08–18.51) 13.45 (11.01–15.47) 12.02–14.70	8.32 ± 1.50 (5.73–11.26) 8.60 (7.37–9.47) 7.49–9.15	0.002
IL-6	13.06 ± 20.16 (3.55–80.25) 5.4 (4.47–7.04) 1.90–24.22	4.99 + 5.00 (2.42–18.27) 2.96 (2.75–4.12) 2.22–7.75	0.11
IFN-gamma	8.84 ± 5.83 (4.70–24.02) 6.41 (6.16–8.33) (5.61–12.07)	5.95 ± 1.02 (4.83–9.37) 5.80 (5.39–5.98) (5.382–6.516)	0.15

OPG, osteoprotegerin; TNF, alpha-tumor necrosis factor alpha; IL, 6-interleukin 6; IFN, gamma-interferon gamma; T0, before treatment; T2, two months after fixing the brackets. Indicators were statistically significantly different when *p* < 0.05.

## Data Availability

Not applicable.

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
