# Peer review of "Influence of Different Orthodontic Brackets on Cytokine and Cortisol Profile"

_medicina, 2023, doi:10.3390/medicina59030566_

Round 1

Reviewer 1 Report

I have read the manuscript “Cytokines as biomarkers of different orthodontic treatments”

The title does not fit the aim and the results. This is a preliminary study and the investigation of concentrations of cytokines (IL-6, 15 osteoprotegerin, TNF-alfa, and IFN gamma) and cortisol was performed according to the type of treatment. To find an association is not enough to report it as a biomarker.

Abstract: the first time the osteoprotegerin appears, should be clear that this is OPG.

Some number would be helpful in the result section of the abstract.

Was the saliva collected in the first week of the treatment, when exactly?

How was sample selection and patients allocated in each group?

Patients data are missing. Gender and age distribution according to the groups should be included, and a statistical analysis comparing the parameters among groups.

Please, include the exact value instead “NS”.

Table 3 the p-values shows numbers with “,”, correct and replace with “.”

Author Response

We would like to thank you for the constructive and positive evaluation of our paper.  We considered all of their comments  carefully and changed the paper accordingly. A detailed point-by-point answers to the reviewer’s comments are provided below.

I have read the manuscript “Cytokines as biomarkers of different orthodontic treatments”

Answer: Thanks to the reviewer for the comment. We changed the title according to referee’s comment and now is “Influence of Diferrent orthodontic brackets on cytokine and cortizol profile”.

The title does not fit the aim and the results. This is a preliminary study and the investigation of concentrations of cytokines (IL-6, 15 osteoprotegerin, TNF-alfa, and IFN gamma) and cortisol was performed according to the type of treatment. To find an association is not enough to report it as a biomarker.

Abstract: the first time the osteoprotegerin appears, should be clear that this is OPG. Some number would be helpful in the result section of the abstract.

Answer: Thanks to the reviewer for the comment, we have added the numbers and also make explanation of osteoprotegrin when are used first time in the text (Please see p.1, Section “Abstract”)

Was the saliva collected in the first week of the treatment, when exactly?

A: Thanks to the reviewer for the comment. Saliva was collected 30 minutes before starting of fixation procedure.

How was sample selection and patients allocated in each group?

Answer: We thank the referrer for the question. As reviewer noted above, our research is preliminary and observational study. Therefore, we collected saliva according to the criteria described in the article (please see section materials and methods), although the researcher did not divide them into separate groups according some special criteria. We collected saliva without randomization as a result of the routine treatment procedure and then divided into 2 groups according to the respective treatment.

Patients data are missing. Gender and age distribution according to the groups should be included, and a statistical analysis comparing the parameters among groups.

Answer: According to reviewer’s request we add the information about gender and age distribution (please see Table 1 (I and II) and p.5)

Please, include the exact value instead “NS”.

Answer: According to reviewer’s request we add exact p values (please see Table 1-3).

Table 3 the p-values shows numbers with “,”, correct and replace with “.”

Answer: According to reviewer’s request we replace in the Table 3 comma with point (Please see, p.5-7 Table 1-3).

Reviewer 2 Report

-        In the abstract, the p-value and the statistical test used should be mentioned.

-        On line 38, identify the acronym PDL before mentioning it.

-        Specify what type of study it is in the methodology; it is only referenced in the end of the article in the limitations.

-        Number the different parts of the article.

-        In the materials and methods section, the inclusion and exclusion criteria should be clarified.

-        Explain how the saliva collection was performed, and if this and the clinical assessment of oral cavity were performed by only one or more investigators.

-        The patients' initial occlusal change should be characterized, for example, the degree of crowding, was it similar? What were the selection criteria for the type of brackets (Conventional or Self-ligating brackets) to use in each individual?

-        The number of individuals included should be justified, as well as mentioning why 10 patients were excluded from the final data.

-        The collection time after treatment (2 months) should be justified.

-        Statistical p must be written in lower case.

-        Throughout the text, traditional, conventional, and standard brackets are mentioned. The nomenclature should be standardized.

-        The tables are incomplete, namely, the confidence interval should be given for the difference and the value of p, even if it is not statistically significant, should be referred to.

-        The results are confusing. When mentioning a table or a graph, perform a complete analysis in its first reference.

-        The information in lines 144 and 145 is not consistent with the results.

-        The justification for the study design must be mentioned in the methods or discussion section and not in the results (line 193 and 194).

-        In the Discussion, the clinical translation of the results found should be further explored.

-        In the limitations, the difference in the value of n between groups should be included.

-        Standardize the way the page numbering is placed in bibliographic references (228-34 or 228-234)

Author Response

We would like to thank you for the constructive and positive evaluation of our paper.  We considered all of their comments  carefully and changed the paper accordingly. A detailed point-by-point answers to the reviewer’s comments are provided below.

This is an interesting article about biomarkers during orthodontic treatment. However, statistical tests are flawed, namely, the distribution of sub-points is irregular, so non-parametric tests should be used, and all results should be presented as medians and interquartile ranges, minimum and maximum).

Answer: The reviewer is correct and we have added the non-parametric test results. However, I must clarify that the graphs show logarithmic indicators (please see table 2 and 3, graphs).

Please put everything in two decimal places.

Answer: We agree with the reviewer and now all numerical values are displayed as requested (see table 1-3).

I recommend reviewing the English language once again because there are grammatical errors throughout the text.

Answer: At the reviewer's recommendation, the MS has been checked and edited by our university English editor, and we now think all grammatical errors have been corrected.

Reviewer 3 Report

This is an interesting article about biomarkers during orthodontic treatment.

however, statistical tests are flawed,

namely, the distribution of sub-points is irregular, so non-parametric tests should be used, and all results should be presented as medians and interquartile ranges, minimum and maximum).

Please put everything in two decimal places.

I recommend reviewing the English language once again because there are grammatical errors throughout the text.

Author Response

We would like to thank you for the constructive and positive evaluation of our paper.  We considered all of their comments  carefully and changed the paper accordingly. A detailed point-by-point answers to the reviewer’s comments are provided below.

  • In the abstract, the p-value and the statistical test used should be mentioned.

Answer: Thanks to the reviewer for the comment, we have added the numbers and statistical p values in the abstract (Please see p.1, Section “Abstract”)

  • On line 38, identify the acronym PDL before mentioning it.

Answer: According to reviewer’s request the acronym is identified (please see p. 1)

  • Specify what type of study it is in the methodology; it is only referenced in the end of the article in the limitations.

Answer: According to reviewer’s request the type of study (observational) is included in the section “subjects”.  (please see p. 3)

  • Number the different parts of the article.

Answer: According to reviewer’s comment the different part is numbered (please see title of the parts: Introduction, Materials and Methods etc.)

  • In the materials and methods section, the inclusion and exclusion criteria should be clarified.

Answer: According to reviewer’s comment inclusion and exclusion criteria are mentioned in the section “Materials and Methods” (please see 2. subjects)

  • Explain how the saliva collection was performed, and if this and the clinical assessment of oral cavity were performed by only one or more investigators.

Answer:  Thanks to the reviewer for the comment. Saliva collection and oral cavity evaluation procedure are explained 2.2 and 2.3 section. Oral cavity clinical evaluation was performed only by one PhD student (MD) and supervisor  (Professor, MD, PhD);

  • The patients' initial occlusal change should be characterized, for example, the degree of crowding, was it similar? What were the selection criteria for the type of brackets (Conventional or Self-ligating brackets) to use in each individual?

Answer: Thank the referrer for the question. Degree of crowding was similar, in all cases it was moderate crowding. As we note before, our research is preliminary and observational study, non-randomized. At the treatment period, the investigator did not determine some special selection criteria for the type of brackets. The purpose of the research is to detect changes in cytokines involved in bone metabolism during the use of different treatment methods in a real situation. Thus, we collected saliva without randomization as a result of the routine treatment procedure.

  • The number of individuals included should be justified, as well as mentioning why 10 patients were excluded from the final data.

Answer: According to reviewer’s comment the number of individuals included in the “materials and methods” section and now it can read as “...Information about 10 patients were withdrawing from the final data because of acute respiratory diseases (Covid-19, flu and etc) during observation period” (please see p.5).

  • The collection time after treatment (2 months) should be justified.

Answer: Thanks for your comment. According to various studies, 6-12 weeks after the start of orthodontic therapy, the most intensive period of growth of microbial and non-microbial indicators is recorded. Thus, 2 months was chosen as the optimal period to detect preliminary changes in cytokines and cortisol.

Alkhayyat DH, Alshammery DA. Real time polymerase chain reaction analysis in the patients treated with fixed appliances after the orthodontic treatment: A follow-up study. Saudi J Biol Sci. 2021 Nov;28(11):6266-627

  • Statistical p must be written in lower case.

Answer: According to reviewer’s comment statistical p values is written in lower case (please see results section)

  • Throughout the text, traditional, conventional, and standard brackets are mentioned. The nomenclature should be standardized.

Answer:According to reviewer’s comment we  standartized nomenclature and use only “conventional brackets” (please see MS).

  • The tables are incomplete, namely, the confidence interval should be given for the difference and the value of p, even if it is not statistically significant, should be referred to.

Answer: According to reviewer’s comment the medians and interquartile ranges, minimum and maximum, also exact p-values were added to the tables (please see Table 1-3)

  • The results are confusing. When mentioning a table or a graph, perform a complete analysis in its first reference.

Answer: According to reviewer’s comment the results section was modified and we mention only table or graph (please see section results, table 1-3).

  • The information in lines 144 and 145 is not consistent with the results.

Answer: According to reviewer’s comment the statitical analysis and results section was modified (please see 2.5 section).

  • The justification for the study design must be mentioned in the methods or discussion section and not in the results (line 193 and 194).

 Answer: According to reviewer’s comment the justification was moved to section “Materials and methods” (please see subsection 2.1 subjects, p. 4).

  • In the Discussion, the clinical translation of the results found should be further explored.

Answer: According to reviewer’s comment in the discussion section was add more explanation about clinical translation (please see discussion, p.10).

  • In the limitations, the difference in the value of n between groups should be included.

Answer: According to reviewer’s comment  in the in the limitations, the difference in the value of n between groups was included (please see p. 10).

  • Standardize the way the page numbering is placed in bibliographic references (228-34 or 228-234)

According to reviewer’s comment the page numbering was changed  (please see section “references”).

Round 2

Reviewer 3 Report

After rewison, the manuscript is improved and it is ready for publication.